# Training Models 20X Faster in Medical Image Analysis

**Author Name1**[1]                                                                                    ABC@SAMPLE.EDU

[1] *Address 1*

## Abstract

Analyzing high-dimensional medical images (2D/3D/4D CT, MRI, large-scale histopathological images, etc.) plays an important role in many biomedical applications, such as anatomical pattern understanding, disease diagnosis, and treatment planning. The AI assisted models have been widely adopted in the domain of medical image analysis with great successes. However, training such models with large-size data is expensive in terms of computation and memory consumption. In this work, we provide solutions for improving model training efficiency, which will speed up the training of AI models (20 times faster on an exemplary 3D segmentation framework), and enable researchers and radiologists to improve the efficiency in their clinical studies. The overall efficiency improvement comes from both improved algorithms and engineering advance.

**Keywords:** Medical image analysis, deep learning, segmentation.

## 1. Introduction

Analyzing high-dimensional medical imaging (2D/3D/4D CT, MRI, large-scale histopathological images, etc.) is becoming increasingly common, and plays an important role in many biomedical applications such as anatomical pattern understanding, disease diagnosis and treatment planning. To better study the medical images, machine learning methods are widely applied to analyze the region-of-interest based on contextual information. Leveraging the large datasets with expert-level annotation, such methods improve the performance in many challenging applications compared to conventional methods.

Deep neural network (DNN) is a promising approach for machine learning and is also used as a means of extracting features to handle low-level vision problems. The main advantage of DNN is that training and inference are automated with very few human heuristics and interaction. Meanwhile, using the modern computing devices (e.g. GPU) makes the training and inference extremely efficient. Nowadays, convolutional neural networks (CNN) is the state-of-the-art method for many applications in medical imaging (Isensee et al., 2018). Although the system with neural networks is relatively straightforward, the question remains that how the networks can be trained properly and efficiently for various applications. Given the large size and computation requirement of the medical imaging, careful designation of all components and their connecting functions is in a stringent need to improve the efficiency and productivity of researchers (Ioannou et al., 2019).

In this paper, we treat neural network training in medical imaging as a challenging problem. As an example, applications in 3D medical image segmentation are utilized because they are typical in medical image analysis and challenging with high resolution and large

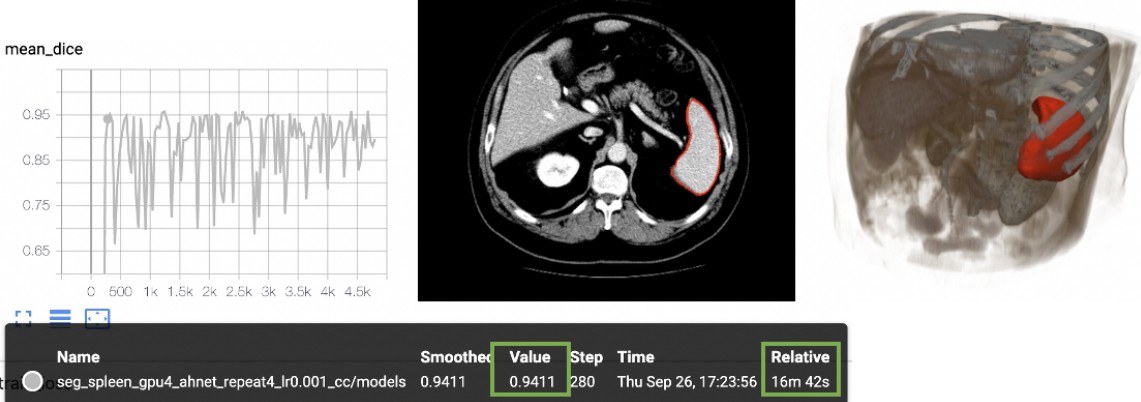

Figure 1: Left: validation accuracy in TensorBoard. Using the proposed framework, the expected validation accuracy 0.940 (Dice's score) is achieved within 16.5 minutes; middle: the red contour denotes prediction from model within one sectional plane; right: volumetric rendering of prediction in 3D space.

data size in 3D. The analysis of the applications proves the effectiveness and efficiency of our proposed framework. The exemplary pipeline will be released soon.

## 2. Background & motivation

In the past decade, numerous deep learning algorithms have been developed for medical imaging analysis, such as segmentation, detection, classifications, etc. Most approaches follow similar pipelines of the corresponding applications in image processing, which mainly apply to 2D images. They usually follow a common basic training pipeline: 1) data feeding, 2) pre-processing and augmentation (e.g. flip, rotate, normalization), 3) sending data to network for prediction, and 4) loss computation and back-propagation. However, the impact of the difference between natural images and medical images on training efficiency has not been studied well in most algorithms. For example, most 2D images has relatively small size, while 3D images are typically much larger: one $512 \times 512 \times 400$ CT image has more than 400MB disk size while one $256 \times 256$ image in ImageNet is only $\sim 250$kB. The I/O loading speed and pre-processing speed that has less effect in natural images processing will trigger the efficiency issue when applied directly to medical imaging use cases.

NiftyNet (Pawlowski et al., 2017) and DLTK (Gibson et al., 2018) are currently two available deep learning toolkits based on TensorFlow (Abadi et al., 2016) for specific applications in medical imaging. They both include various modules designed for medical imaging analysis with deep learning, such as data loader, transforms, networks, losses, and metrics. They also provide multiple examples for medical imaging segmentation, classification and image synthesis. However, they focus more on the correctness of algorithm in development, while paying less attention to the training efficiency, which shall hinder their applications in large scale training tasks. Take GPU cluster use cases as an example, if most jobs have low efficiency and require long training times, less jobs will be finished

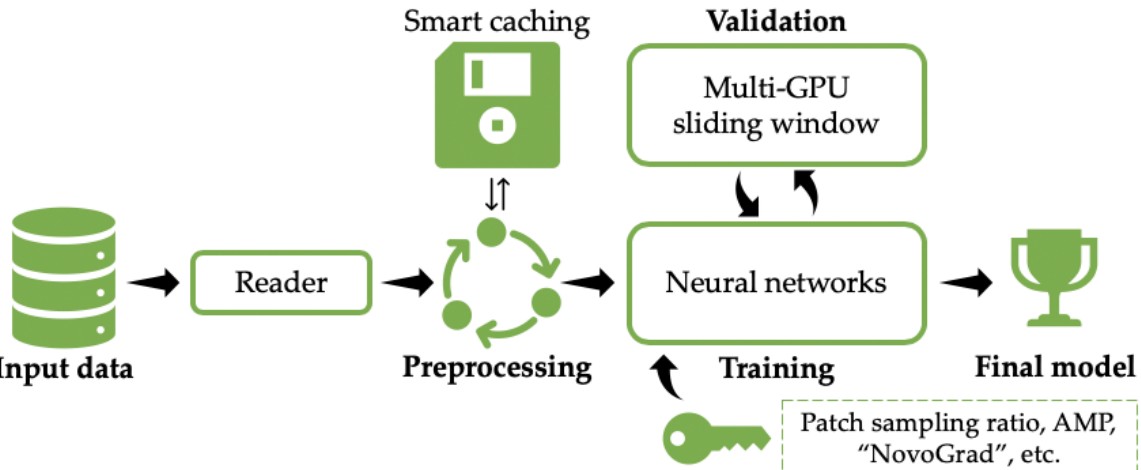

Figure 2: The flowchart of our proposed framework.

within the same amount of time, which results in less productivity for researchers. In the meantime, running experiments with multiple parameter combinations is usually necessary to obtain the best configuration. This becomes problematic when applying to large number of training data, which tends to result in more robust and powerful model. In this paper, we developed multiple efficient modules for medical imaging analysis, which greatly reduces the training time and improves the GPU utilization rate.

## 3. Execution model

In the modern medical image processing, segmentation is one of the most important tasks for various applications, including clinical study, disease diagnosis, and treatment planning. During the segmentation process, the anatomical objects (e.g. organs, bones, or tumors) are identified and parsed from 2D, 3D, or 4D medical images via automated or semi-automated approaches. How to process the large-size data (e.g. 3D/4D CT, MRI) with high resolution in an accurate yet efficient fashion has been challenging for the image segmentation. Computing large-volume data can cause massive GPU memory consumption and long computation time, even for simple convolution operations. This issue becomes even more serious as the complexity of operations or the quantity of data points increases. Therefore, how to improve the model training efficiency has attracted great interests from both academia and industry. In this work, we propose an effective solution to improve the model training efficiency by a large scale (20X faster), leveraging the recent advances from both algorithmic and engineering sides.

To validate our proposed work, we use the organ and lesion segmentation in 3D abdominal CT and cardiac MRI for demonstration. Similar approaches can be applied to other related tasks, such as 3D/4D medical image classification, detection and segmentation. The details of our findings are shown in the following section. The model training time has been successfully reduced from $5 - 8$ hours (the baseline version) to less than 20 minutes.

## 4. Extensibility case studies

**Datasets** The Medical Segmentation Decathlon (MSD) includes several tasks of 3D medical image segmentation (msd, 2018). We choose the tasks of spleen CT segmentation, left atrium MRI segmentation, liver and liver lesion CT segmentation for experiments. Three tasks covers different imaging modalities and different types of body anatomy (large organs versus small lesions). And we split the data for training/validation (specifically, for spleen segmentation, 32 CT volumes for training, and 9 CT volumes for validation; for left atrium segmentation, 16 MRI volumes for training, and 4 volumes for validation; for liver and lesion segmentation, 104 CT volumes for training, and 27 volumes for validation). Datasets are re-sampled into the isotropic resolution 1.0 mm to ensure that fine details would be well observed by models, and model has balanced reception in 3D space. For CT volumes, The voxel intensities of the images are normalized to the range according to the 5th and 95th percentiles of the overall foreground intensities. For MRI volumes, the non-zero voxel intensities are processed via standard normalization. The largest image size after re-sampling in the training pool is [462, 462, 520]. Because of its size, the entire volume cannot be fitted into any the "state-of-the-art" segmentation network directly on a single GPU for either training or inference.

**Baseline approach** The baseline approach is straight forward using fully convolutional network, which has won the 2$^{\text{nd}}$ place of the MSD challenge in 2018. In this approach, the network input is the patches with the size of [112, 112, 112] during training, which are randomly cropped from 3D volumes. Relevant data augmentation techniques, including random axis flipping and random intensity shift, are used for training models. Moreover, the validation follows the scanning-window fashion with small overlaps (32 voxels) between neighboring windows. The window size is [224, 224, 224]. The scanning-window scheme is used mainly to ensure that the input size would not run out-of-memory during computation. The validation accuracy is measured with per-class Dice's score. Accordingly, we use soft dice loss (Milletari et al., 2016) as optimization objectives for all our experiments. $\hat{p}$ and $g$ are model output (before thresholding) and ground truth, respectively.

$$l_{dice} = 1 - \frac{2 \sum_{i=1}^{N} \hat{p}_i \cdot g_i}{\sum_{i=1}^{N} \hat{p}_i^2 + \sum_{i=1}^{N} g_i^2} \tag{1}$$

For the baseline approach, the optimizer is Adam optimizer with learning rate 0.0001. The whole framework is implemented with TensorFlow v1.x and trained on NVIDIA V100 GPUs with 16 GB memory.

The expected Dice's score for the spleen segmentation is around 0.940, 0.912 for left atrium segmentation in MRI. The expected numbers for liver and lesion segmentation are 0.940 and 0.520, respectively. As far as we know, the aforementioned Dice's scores are the state-of-the-art performance given contemporary studies. Our experiments focus more on model convergence, and how and when our model reaches the target performance. Thus, we intend not to train models for a long time.

## 5. Implementation and evaluation

**Neural network architectures** The established network architectures, 3D U-Net (Ronneberger et al., 2015), is used for investigation. The network follows the fashion of con-

| Tasks | Spleen | | Left atrium | | Liver & lesion | | |
|---|---|---|---|---|---|---|---|
| Best metrics | Acc. | Time | Acc. | Time | Acc. 1 | Acc. 2 | Time |
| Native TF | 0.940 | 13h | 0.912 | 10h | 0.940 | 0.520 | 86h |
| Proposed | **0.948** | **30m** | **0.917** | **16m** | **0.941** | **0.545** | **7h** |

Table 1: Comparison on three tasks with single GPU. The numbers are the best accuracy at the specific time. Our proposed framework has clear advantage in terms of performance and efficiency.

| Num. of GPUs | 1 | 4 | 8 |
|---|---|---|---|
| Native TF | 13h | 9.5h | 7.5h |
| Proposed | **30m** | **16.5m** | **8m** |

Table 2: Comparison on Spleen CT segmentation with different number of GPUs. More GPUs not only increase batch size for training, but also reduce on-the-fly validation time.

| Tasks | Spleen | | Liver & lesion | |
|---|---|---|---|---|
| Metrics | Acc. | Iter. | Acc. | Iter. |
| Non Ada. | **0.956** | 1.44k | 0.733 | 13.4k |
| Ada. | 0.952 | **1.28k** | **0.735** | **8.4k** |

Table 3: Validation on adaptive patch sampling strategy (Ada.). With adaptive strategy, the models achieved the expected performance with much less training iterations.

volutional encoder-decoder. We also experimented the vanillar U-Net with other types of normalization methods to replace batch normalization. Because batch size of 3D networks cannot be large due to GPU memory limitation, and batch normalization works less stable with small batch size.

**Smart caching and smarter caching** It requires efficient use of computing resources and effective use of training data to train DNN models with high speed and descent accuracy. The major computing resources are IO, CPU, and GPU, where IO is usually the bottleneck and GPU would be the most efficient. Though GPUs provides efficient computing, it is not a general-purpose computing device. Execution of training algorithm is done by CPU.

In model training, the same training dataset is used repeatedly. In each epoch of the training, a set of training images must be loaded from files and then processed through a chain of transformations, before fed to the training graph for GPU accelerated computation. Transformation represents data pre-processing and augmentation. Currently all transformations take place on CPU. IO and data transformation are the major bottleneck of training efficiency, causing low GPU utilization rate. The problem becomes even severe with 3D datasets and 3D networks. Think about the time needed to load multiple 3D volumes each with hundreds of MB in every training iteration. We developed the smart caching technique to store some intermediate results in the RAM which leverage the high communication speed between GPU and RAM, reducing the burden on slow IO. It reduces the time that GPU is waiting for each batch data to be ready for training, increases the GPU utilization rate and reduces the training time.

The 1$^{st}$ smart thing of smart cache is determining the most effective data to **cache based on the determinism of transformations**. By most effective, we mean "generating

most of time saving". Note that data loading is also considered a data transformation. A transformation is called deterministic if it always produces the same output for the same input. Data loading is deterministic, and so are many other transforms. Another observation is that the sequence of multiple deterministic transforms is also deterministic. Given a chain of transforms, the most effective data to cache is the output of the longest deterministic sequence of transforms, starting from the 1st transform. By caching this data, we bypass all transforms in the sequence in future uses.

The training dataset can be arbitrarily large; hence we cannot assume that all data items can be cached in memory at the same time, especially when the data items are large 3D volumes. Therefore, we must allow the cache capacity to be configured. Then we have a problem. As we know, all data items are repeatedly used with equal chance. If the caching space is not big enough to hold all data items, the items not in cache will still be subject to the slowness of disk IO and transformation. This is where the smart cache offers the 2nd smart thing: **gradual replacement of cache contents**.

It works as follows. First of all, the smart cache must be configured properly: 1) number of cached objects: the number of objects to be cached; 2) replacing rate: percentage of objects to be replaced in each round. At any time, the cache pool only keeps a subset of the whole. In each round (epoch), only the items in the cache pool are used for training. This ensures that the data items needed for training are readily available, hence keeping GPU busy. Note that cached items may still have to go through some non-deterministic transformations before fed to GPU. At the same time, another thread is preparing replacement items by applying the deterministic transform sequence to selected items not in cache. Once one round is completed, smart cache replaces the same number of items with the replacement items.

The smart caching covers deterministic components of computation chain (data pre-processing) without any requirement of disk space. After applying the techniques, the training time is reduced by 50%, and the GPU utilization rate maintains at an extremely high level (close to 100%).

**Adaptive positive/negative sampling ratio** Because the sizes of training data vary resulted from the different field-of-views and GPU memory limit does not allow training networks with the entire 3D volumes, image cropping is necessary during training. During each iteration, the volumetric patches are sampled from different data points to form the mini-batch. Due to highly imbalance issue of foreground and background, the patches needs to be carefully sampled for each iteration.

We define two kinds of patches: positive patches $P_+$ and negative patches $P_-$. $P_+$ is defined as its center voxel represents foreground classes in ground truth label, and $P_-$'s center voxel represents background classes. A straightforward strategy would be sampled $P_+$ and $P_-$ with equal chances to train networks with balanced data distribution. The strategy works well but takes long time to ensure model convergence, because models need to see enough background samples to perform robustly (here we assume background is the majority class). Another strategy could be sampling $P_+$ and $P_-$ based on overall class distribution in terms of voxel quantities. However, the network trained with large amount of $P_-$ and the false negative rate in the final prediction would raise.

In order to reach a balanced but efficient patch sampling strategy, we proposed an adaptive method to adjust ratio $r$ between $P_+$ and $P_-$ dynamically during training. Let's

set $r = 1:1$ in the beginning of training. After first-time validation, we can compute the false positive index $r_{\text{FP}}$ and false negative index $r_{\text{FN}}$ for the prediction as follows.

$$r_{\text{FP}} = \frac{1}{C} \cdot \sum_c \frac{\sum_n p_{n,c} \cdot (1 - g_{n,c})}{\sum_n p_{n,c} + g_{n,c} - 2 \cdot p_{n,c} \cdot + g_{n,c}} \tag{2}$$

$$r_{\text{FN}} = \frac{1}{C} \cdot \sum_c \frac{\sum_n (1 - p_{n,c}) \cdot g_{n,c}}{\sum_n p_{n,c} + g_{n,c} - 2 \cdot p_{n,c} \cdot + g_{n,c}} \tag{3}$$

$r_{\text{FP}}$ is defined as the portion of false positive error with the whole error mass, and $r_{\text{FN}}$ is defined as the portion of false negative error. In Eq. 3, $p_c$ and $g_c$ denotes prediction (after thresholding) and ground truth for foreground class $c$. $C$ is the number of foreground classes, and $n$ is the voxel location. Once $r_{\text{FP}}$ and $r_{\text{FN}}$ are computed with validation ground truth, the new ratio $r'$ between positive and negative patches for next epoch can be determined as follows. $\gamma$ is a tunable parameter with positive value, which controls the magnitude of ratio changes for next epoch of training. $\gamma$ is set to 1 in our experiments.

$$r' = \left( \frac{r_{\text{FN}}}{r_{\text{FP}}} \right)^{\gamma} \tag{4}$$

For instance, larger $r_{\text{FP}}$ means more false positive errors within prediction. Then the model should see more negative samples in the future training. We update $r$ after each validation during training, the model would be trained with balanced samples and training efficiency is largely improved.

However, the adaptive sampling strategy could be hazard when model training is almost converged. At that time, both types of error are very small but the ratio $r_{\text{FP}}$ and $r_{\text{FN}}$ could be very large. Large but imbalanced $r$ would slow down the convergence, make the training process unstable. Thus, we make further improvement based on Eq. 3 as following equations.

$$r_{\text{FP}} = \frac{1}{C} \cdot \sum_c \frac{\sum_n p_{n,c} \cdot (1 - g_{n,c}) + p_{n,c} \cdot g_{n,c}}{\sum_n p_{n,c} + g_{n,c} - p_{n,c} \cdot g_{n,c}} = \frac{1}{C} \cdot \sum_c \frac{\sum_n p_{n,c}}{\sum_n p_{n,c} + g_{n,c} - p_{n,c} \cdot g_{n,c}} \tag{5}$$

$$r_{\text{FN}} = \frac{1}{C} \cdot \sum_c \frac{\sum_n (1 - p_{n,c}) \cdot g_{n,c} + p_{n,c} \cdot g_{n,c}}{\sum_n p_{n,c} + g_{n,c} - p_{n,c} \cdot g_{n,c}} = \frac{1}{C} \cdot \sum_c \frac{\sum_n g_{n,c}}{\sum_n p_{n,c} + g_{n,c} - p_{n,c} \cdot g_{n,c}} \tag{6}$$

We add true positive prediction to both numerator and denominator. The new definitions represent relationship between false positives and false negatives. And when errors become very small, $r_{\text{FP}}$ and $r_{\text{FN}}$ are all close to 1. The balanced sampling helps for model convergence at the end of training.

The effectiveness of the proposed patch sampling strategy is shown in Table 3. With adaptive strategy, the models achieved the expected performance of different applications with much less training iterations.

**Optimizer** The novel "NovoGrad" optimizer utilizes the layer-wise gradient normalization together with stochastic gradient descent (SGD), which has been validated in several applications, such as ImageNet classification, and applications of natural language processing (Ginsburg et al., 2019). "NovoGrad" works better with faster convergence rate, compared to Adam or vanilla SGD.

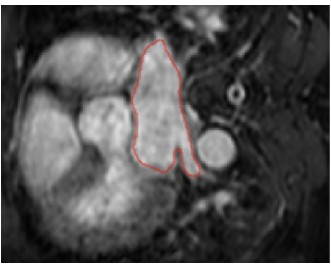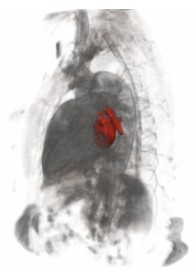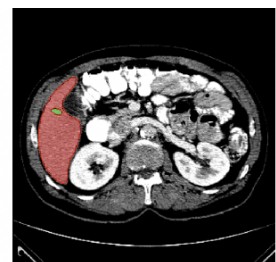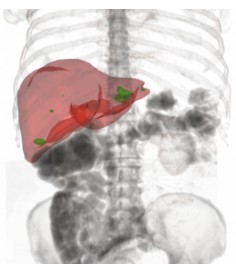

Figure 3: Left: the red contour denotes LA prediction from model, and volumetric rendering in space (model trained within 6 minutes); right: the red and green contours denote prediction of liver and lesion respectively, and volumetric rendering in space (model trained within 6 hours).

**Automated Mixed Precision (AMP)** AMP is an advanced technology to reduce training time and memory consumption (Micikevicius et al., 2017). The main idea is that converting part of operations into float16 operations during training in order to reduce resource demands. Meanwhile, the training efficiency maintains at the same level.

**Determinism** Our framework enable the deterministic training with TensorFlow following instructions in (det). After setting all possible random seeds (Python, TensorFlow, Numpy etc.), the results are fully reproducible after training with the same computing environment. The random seeds are the same across all experiments to make fair comparison (models see the same patches with the order during training).

**Multi-GPU** We conducts experiments with 1 GPU, 4 GPUs, and 8 GPUs shown in Table 2. More GPUs for training, enabled by (hor), mean larger batch size, and it does improve the convergence rate. The bonus to use multi-GPU is enabling efficient validation, which means different GPU works on separate data points with sliding-window scheme.

**In general, the most efficient and effective framework is the combination of smart caching, adaptive patch sampling, AMP, multi-GPU, and "NovoGrad".** Our proposed frame has outstanding efficiency and effectiveness with different image modalities and human anatomies (shown in Table 1). The conventional U-Net models can be trained in an efficient fashion. In the Table 1. The "Native TF" denotes one of the available public toolkits with TensorFlow for medical image analysis. The liver lesion segmentation takes much longer time because the dataset is relatively large (compared to other two), and liver lesion segmentation is challenging (small lesion with large background area). Our framework is capable to train conventional network more than 20 times faster with state-of-the-art accuracy.

## 6. Conclusions

To summarize, the optimal training solution is a combination of several advanced technologies. Multi-GPU training using smart caching, adaptive sampling, AMP and "NovoGrad" introduces promising training efficiency improvements. By carefully designing for each specific application, model training can be increased more than 20 times faster for large-scale medical image analysis. We hope that a wide audience, including but not limited to researchers and radiologists, can benefit from our proposed solutions.

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

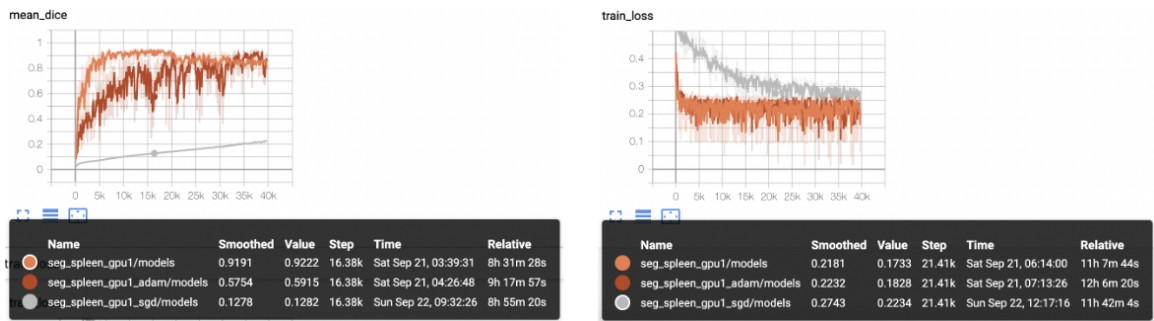

Figure 4: Left: validation accuracy for different optimizer ("NovoGrad" is the orange curve); right: training loss for different optimizer. Apparently, "NovoGrad" works better with faster convergence rate, compared to Adam or vanilla SGD.

## Appendix A. More experiments

**Post-processing** "Keeping the largest connected component" would introduce additional benefits to improve the model training efficiency for organ segmentation. And the training time for organ segmentation can be further reduced.

**AMP** AMP is able to achieve the same level of validation accuracy, compared to training without AMP, shown in Figure 5. And the average validation accuracy over all epochs has a little gap shown in the figure. This would be the side effect when converting some of operations into half precision format (e.g. float16 casting, loss scaling). However, it does not matter if we care more about the best validation scores.

The speedup is mostly dependent on how many nodes in the graph can be converted into the half precision format. It is fully determined be the AMP algorithm. In the case shown in the figure, only 10% of nodes are able to be converted. In general, 3D conv. was not well supported until the most recent TensorFlow.

**Misc.** We run experiments on several normalization options: batch normalization, instance normalization, layer normalization, weight normalization. Instance normalization is computed across different instances, layer normalization is computed for each layer, and weight normalization is conducted on the kernel weights. U-Net is used here for comparison. Weight normalization does not help for convergence, so its result can be ignored. The light blue curve represents the result of using instance normalization, which works the best in this scenario. The batch normalization causes the slow convergence since our batch size is small compared to the one used in 2D image classification. The statistics of a mini-batch cannot fully represent the full volume. (For all our experiments, batch size is 6 per GPU instance shown in Figure 5.)

Batch size options for training: 2, 4, 6 (default), 12. In general, the large batch size works better for convergence, but the side-effect is that the large batch size requires longer time for model training. The trade-off decision needs to be made by users.

Input shape options for training: $64^3$, $112^3$(default), $128^3$ shown in Figure 6. In general, the large patch size works better for convergence, but the side-effect is that large patch size

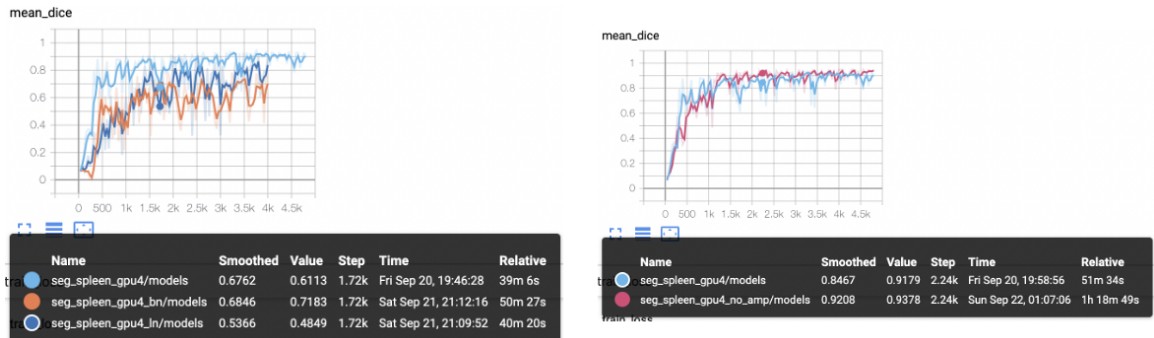

Figure 5: Left: validation accuracy for three normalization layers, instance normalization (IN), batch normalization (BN), and layer normalization (LN). Instance normalization is the light blue curve; right: validation accuracy for the same the framework with or without AMP. Using AMP is able to maintain the same level of accuracy but with higher efficiency.

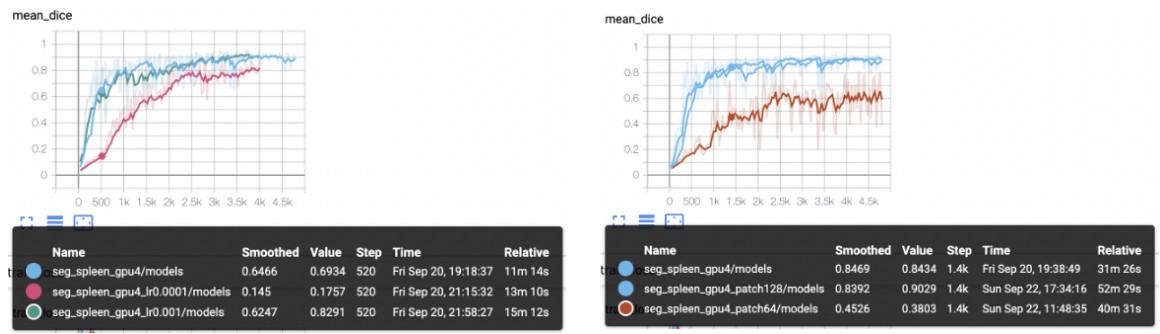

Figure 6: Left: validation accuracy for three different learning rate. Learning rate 0.0005 is the light blue curve; right: validation accuracy for different patch sizes (64, 112, 128).

requires longer time for model training. The trade-off decision needs to be made by users. However, the patch size cannot be too small.

Learning rate options: 0.001, 0.0005 (default), 0.0001. Large learning rate promotes the fast convergence shown in Figure 6, but the validation score will be less stable compared to the results from smaller learning rates.

