# OpenReview forum: "Training Models 20X Faster in Medical Image Analysis"
_MIDL.io/2020/Conference — Submitted to MIDL 2020_

### Official Review · AnonReviewer2 · 2020-03-12
**Review on the paper**

**Rating:** 3
**Confidence:** 4
**Recommendation:** Poster

**Summary:**

In this paper, the authors try to overcome the issue over long training times needed for training deep learning models in medical image analysis. The authors focus on a 3D-segmentation task that is known to be memory intensive for both GPU and CPU. They suggest a smarter caching method that keeps track of the false positive and false negatives. They compare their results to a baseline without a smarter cache.

**Strengths:**

Overall the paper is well written and nicely structured. The methods are clearly described. The authors focus on reducing training time for researchers in the field of medical image analysis with the possibility to scale further.

**Weaknesses:**

Some areas need some clarification, see below.
- Could the authors give some insight how much time is spent on loading images from disk, augmentations, CPU time, etc.
- Could the authors give some details on what kind of system they are using, what kind of CPU, what kind of HDD/SSD, how much memory etc. This can influence the amount of disk IO.
- The smart caching method keeps track of the FP, FN, TP and TN's after a validation run. How does this compare to on the fly hard negative mining?
- Can the authors explain how this would scale for a multi-class problem.
- Could the authors compare their results to a network where i.e. only AMP or NovoGrad is applied.
- Could the authors explain how the increased number of GPU's ensures that there is less disk IO. Since there are more GPU's that need to be fed with patches, and therefore increasing memory footprint.
- Minor: the references to tables and figures are inconsistent. Some tables/figures are never referenced and the order is also not always correct. Please address.

**Justification Of Rating:**

The paper discusses an important item that all researchers in the field however, some critical points are not clearly described in the paper. It's not clear on what kind of system the authors tested the solution, it's not clear how the technique differs from hard negative mining.

**Paper Type:**

validation/application paper

**Special Issue:**

no

---

### Official Review · AnonReviewer3 · 2020-03-13
**Training models x20 faster**

**Rating:** 2
**Confidence:** 4

**Summary:**

The paper presents a framework to speed up computation time and memory consumption of machine learning models in the medical image domain. Medical images and scans are large in data due to their rich nature, so problems in the field arise from not being able to fit the entire network on a GPU, slow data reads, and not fully exploiting cache. The authors propose smart caching, saving intermediate results in the RAM, reduction of training data to 16 bit floats, using multi GPUs, using an optimized tailored for fast convergence in order to achieve a training late of overall 20 times faster than before.

**Strengths:**

Main contribution/strength of the paper is to achieve substantial improvements in the training times for large scale medical image models. By exploiting a myriad of fine techniques at different steps of the entire system, overall speed up, which is 20 times improvement in the speed, is substantial and offers a new venue for  the adoption of their frameworks in the field by anyone who is dealing with large scale medical image dataset/system. Other strength of the paper is that it plays the role of a handbook for anyone who is interested in improving their AI models without undergoing a complete change.One can read this paper and adapt one or some of the steps as guidelines in order to speed up their process, instead of fully rewriting their code base, or changing their infrastructure that they used for their research.

**Weaknesses:**

This paper is being evaluated on the improved methodology category even though some portion of the steps that the authors took fall into validation because for some steps, they simply reuse existing improvements noted in the literature again.

That being said, some optimization techniques utilized by the authors are as simple as changing the bit size during training, or changing the optimizer, or changing the GPU count.

 In terms of a validation category, this paper reports impressive results since they are definitely a sign of improvement in the computational running time. But usually, for a framework like this, if there is a speed up in one section, there is a slow down in other sections. I would be interested in learning how much of an overhead their preparation took when they wanted to apply their framework.

Overall, the results show improvements since one can assume that the overall cost of adapting this framework from scratch should not be more than the initial cost of training the baseline AI model, let’s say Native TF , which was 13 hours in for Spleen scans for one their cases.



**Justification Of Rating:**

This paper has a very practical objective, that is to increase speed ups.
 The authors seem to miss the discussion on the preparation time/ overhead caused by their framework. That is, how much change does one need to do in their current implementation of machine learning model creating process in order to adapt their framework?
Also, the majority of the improved steps in their framework is as simple as changing the GPU count, or changing bitsize of numbers, or changing optimizers, which doesn’t offer a great novel approach rather than being an aggregation of previously reported refinements in the literature.

It is definitely a useful paper, but not sure if this is the right medium for publication.

**Paper Type:**

both

**Special Issue:**

no

---

### Official Review · AnonReviewer4 · 2020-03-14
**Train a faster model  for 3D medical image segmentation**

**Rating:** 2
**Confidence:** 5

**Summary:**

Analyzing high-dimensional medical images (2D/3D/4D CT, MRI, histopathological images, etc.) plays an important role in many biomedical applications. In this paper, the author provides a new solution for improving model training efficiency roughly 20X faster than the original training process. The motivation of this paper is to enable researchers and radiologists to improve efficiency in their clinical studies.

**Strengths:**

Model compression is an important problem in the deep learning domain. There are many works on CNN/RNN compression since 2016. However, there is a lack of work for model compression on the 3D medical image problem. The motivation of this paper is to speed up the training process rather than the inference time or compress the model size, which is very useful for medical image community

The dataset and network configuration were described very clearly.

**Weaknesses:**

1. I would suggest the author shows the comparison snapshot (before compression vs. after compression) to better illustrate the model efficiency.

2. Is the section 3 is the proposed method? and it confused me that seems like a similar description as section 2. I feel like these should all go into the motivation section.

3. The "Smart caching and smarter caching" and "Adaptive positive/negative sampling ratio" should go to the method part, not the experimental settings.

4. I understand the intuition and motivation of this work , however, could the author clearly state what is new here or what is the novelty of this paper? From the current draft, I cannot find what specific new thing in the method.



**Justification Of Rating:**

Based on the current draft, there are many things not clear. For example, the novelty or the ```''new'' idea here and the after reading the whole draft I still not really follow the proposed method. I would give a weak reject. Unless, in the rebuttal, the author can address all my questions.

**Paper Type:**

methodological development

**Questions To Address In The Rebuttal:**

1. I would suggest the author shows the comparison snapshot (before compression vs. after compression) to better illustrate the model efficiency.

2. Is the section 3 is the proposed method? and it confused me that seems like a similar description as section 2. I feel like these should all go into the motivation section.

3. The "Smart caching and smarter caching" and "Adaptive positive/negative sampling ratio" should go to the method part, not the experimental settings.

4. I understand the intuition and motivation of this work, however, could the author clearly state what is new here or what is the novelty of this paper? From the current draft, I cannot find what specific new thing in the method.


**Special Issue:**

no

---

### Official Review · AnonReviewer1 · 2020-03-14
**A bag of computational tricks that are missing links to related work and missing implementation details make comparisons impossible**

**Rating:** 1
**Confidence:** 4

**Summary:**

The authors present computational improvements to speed up training in medical image analysis. They propose a joint system that combines "smart caching", adaptive patch sampling ratios, the NovoGrad optimizer, mixed precision computation and multi-GPU use. The presented results show a 12x - 26x reduction in training times over a baseline. If true, this would be a significant step forward.

**Strengths:**

The main strengths in this paper lie in the detailed description of the proposed method. The schematic figure (figure 2) also helps convey the pipeline strategy. Results are well organized into tables that are easily understood.

**Weaknesses:**

This paper has three main weaknesses: 1) it does not discuss extremely similar ideas 2) the implementation description for the baseline practically none and 3) the manuscript itself is not of publication quality

1) Related work: "smart cache" is presented in this paper as a completely novel idea, but pre-loading data into RAM is not a new concept. For example, TensorFlow has advanced pre-fetching of data (https://www.tensorflow.org/guide/data_performance). Pytorch and other frameworks have similar features. None of these are mentioned at all in the paper and no performance comparison is made

2) The baseline is severely lacking. It is not conveyed what training set-up was used. A number of trivial settings (such as not using built-in data prefetching) can cause training times to increase 20x and therefore make the comparisons meaningless.

3) The paper contains a very high number of grammatical and spelling errors. Screenshots of tensorboard are not publication quality graphics, and Chapter 3 "Execution model" contains no detail on the execution model. Two of frameworks discussed are misattributed. It will require significant rewriting and editing before publication.

**Detailed Comments:**

A sample of minor issues:

- "the system with neural networks is relatively straightforward" needs elaboration
- "networks is" -> "networks are"
- "less jobs" -> "fewer jobs"
- NiftyNet is by Gibson et al. and DLTK is by Pawlowski et al.
- "Smart caching" section includes repeated discussions on data transformation
- 1st -> first
- "GPU memory limit does not allow training networks with the entire 3D volumes" that is entirely dependent on the model and GPU
- There are a number of other grammar faults. Too many to list here.

**Justification Of Rating:**

Although the motivation is clear and reported results claim a significant improvement it is impossible to tell whether the difference stems from any of the methods discussed or from implementation settings for the baseline that cause slow training times. For a computational complexity argument there is a lack of theoretical bounds (even approximations thereof)

Similarly, many of the ideas here (e.g. "smart caching") are very similar to available features (e.g. pre-fetching in tensorflow). The authors have not discussed what the advantages or limitations of these two are.

**Paper Type:**

both

**Questions To Address In The Rebuttal:**

1) What training set-up is used for the baseline? What is "native FT"? Do you prefetch data for the baseline using the built-in tensorboard feature (https://www.tensorflow.org/guide/data_performance)?
2) What is the algorithm used to for "smart caching"? Pseudocode example would be suitable
3) What is the complexity of this algorithm? At what point does the overhead to calculate caching strategy outweigh the time saved?
4) How does your adaptive sampling method compare to other adaptive sampling methods?
5) What is the advantage of each proposed method (in %)?

**Special Issue:**

no

---

### Meta-Review · Area_Chair1 · 2020-04-06
**MetaReview of Paper280 by AreaChair1**

**Rating:** 2

**Metareview:**

All the reviewers agreed that the topic of this paper is interesting however they pointed out a lot of limitations for this paper such as the discussion about the overhead and preparation time of the proposed setup and comparison with other methods similar to the ones suggested by Reviewer 1. The authors did not submit a rebuttal to address the raised concerns, so I agree with the reviewers that the current draft has a lot of things that are unclear that does not make it ready for publication.

**Paper Type:**

both

**Special Issue:**

no

---

### Decision · Program_Chairs · 2020-04-11

Reject